

# A machine learning emulator for Lagrangian particle dispersion model footprints: a case study using NAME

Elena Fillola[1], Raul Santos-Rodriguez[1], Alistair Manning[2], Simon O'Doherty[3], and Matt Rigby[3]

[1]Department of Engineering Mathematics, University of Bristol, Bristol, UK
[2]Hadley Centre, Met Office, Exeter, UK
[3]School of Chemistry, University of Bristol, Bristol, UK

**Correspondence:** Elena Fillola (elena.fillolmayoral@bristol.ac.uk)

**Abstract.** Lagrangian particle dispersion models (LPDMs) have been used extensively to calculate source-receptor relationships ("footprints") for use in applications such as greenhouse gas (GHG) flux inversions. Because a single model simulation is required for each data point, LPDMs do not scale well to applications with large data sets such as flux inversions using satellite observations. Here, we develop a proof-of-concept machine learning emulator for LPDM footprints over a 350 km by 230 km region around an observation point, and test it for a range of in situ measurement sites from around the world. As opposed to previous approaches to footprint approximation, it does not require the interpolation or smoothing of footprints produced by the LPDM. Instead, the footprint is emulated entirely from meteorological inputs. This is achieved by independently emulating the footprint magnitude at each grid cell in the domain using gradient-boosted regression trees (GBRTs) with a selection of meteorological variables as inputs. The emulator is trained based on footprints from the UK Met Office Numerical Atmospheric dispersion Modelling Environment (NAME) for 2014 and 2015, and the emulated footprints are evaluated against hourly NAME output from 2016 and 2020. When compared to $CH_4$ concentration time series generated by NAME, we show that our emulator achieves a mean R-squared score of 0.69 across all sites investigated between 2016 and 2020. The emulator can predict a footprint in around 10 ms, compared to around 10 minutes for the 3D simulator. This simple and interpretable proof-of-concept emulator demonstrates the potential of machine learning for LPDM emulation.

## 1 Introduction

To monitor the efficacy of climate agreements and understand climate feedbacks, there is an urgent need to quantify changing greenhouse gas (GHG) fluxes. Flux inference or inverse modelling systems are becoming increasingly popular for GHG flux quantification, as they produce estimates of the spatial distribution of methane sources from atmospheric observations, using an atmospheric transport model and statistical inversion framework. They have been used, for example, for the evaluation of the UK and Europe's methane emissions using in situ sensors (Lunt et al., 2021; Bergamaschi et al., 2018), the investigation of regional CFC-11 emissions from eastern China (Rigby et al., 2019), and many other applications.

Flux inference inverse methods were traditionally designed for relatively small data sets based on high-precision ground-based measurements. However, the growth of surface networks and space-based observations mean that the volume of GHG data has increased by several orders of magnitude in recent years, and will continue to grow in the next decade. For example,



the TROPOMI instrument onboard the Sentinel 5-precursor, which was launched in 2017, collects around seven million $CH_4$ soundings per day (Butz et al., 2012), compared to ten thousand per day from the GOSAT instrument, launched in 2009 (Taylor et al., 2022). This growth is causing increasingly severe computational bottlenecks for GHG flux inference systems. In particular, systems relying on backward running Lagrangian Particle Dispersion Models (LPDMs) to solve for atmospheric transport are particularly impacted, as the required number of model evaluations grows with number of observations.

Flux inference systems primarily use one of two types of system to simulate atmospheric transport: Lagrangian Particle Dispersion Models (LPDMs), or Eulerian models. LPDMs simulate trace gas transport by following hypothetical "particles" as they move according to 3D "analysis" meteorology provided by forecasting centres. Their main advantage for GHG flux evaluation is that transport can be run backwards in time. This means that the sensitivity of GHG concentration measurements to upwind emissions, often called the *footprint* of an observation, can be calculated directly. This property makes them relatively

simple and flexible to apply to GHG flux evaluation, and, when the number of observations is small, they provide a highly efficient estimate of the sensitivity of those observations to the surrounding high-dimensional flux field. Examples of widely used LPDMs are the Numerical Atmospheric Modelling Environment (NAME, Jones et al., 2007), the Stochastic Time-Inverted Lagrangian Transport Model (STILT, Fasoli et al., 2018), and the FLEXible PARTicle Dispersion Model (FLEXPART, Pisso et al., 2019). Eulerian models, which calculate concentrations throughout a 3D atmospheric grid, do not suffer from the same

scaling problem as the number of observations grows. However, because they do not directly calculate source-receptor relationships, they require the development of complex "adjoint" model codes (e.g. Kaminski et al., 1999), low-resolution finite difference schemes (e.g. Zammit-Mangion et al., 2022), or relatively expensive ensemble simulations (e.g. Peters et al., 2005). If LPDMs are to be used in inverse modelling studies using very large data sets, methods must be developed to overcome their poor scaling with number of observations.

Machine learning has been shown to be useful for efficiently addressing a number of problems in studies using atmospheric dispersion models, including the correction of bias (Ivatt and Evans, 2020) and urban-scale pollution modelling (Mendil et al., 2022). LPDM emulators have been developed to simulate volcanic ash plumes or releases from nuclear plants. For example, Gunawardena et al. (2021) use linear regression to predict footprints for a range of model configurations, Lucas et al. (2017) use Gradient Boosted Regression Trees (GBRTs) to predict outputs for a WRF-FLEXPART ensemble, Francom et al. (2019)

use empirical orthogonal functions to reduce dimensionality and Bayesian adaptive splines to model the plume coefficients for different release characteristics, and Harvey et al. (2018) use polynomial functions to estimate average ash column loads in nearby locations for different model parameters. These studies all have two main factors in common: they all model forward dispersion rather than backwards, and they focus on a single point source and emissions event, looking at the ensemble members produced by different LPDM configurations.

A small number of methods have been developed to efficiently approximate LPDM footprints, mostly using interpolation or smoothing: Fasoli et al. (2018) propose a method to run the LPDM with a small number of particles and use kernel density estimations to infer the the full footprint, Roten et al. (2021) suggest a method to spatially interpolate footprints using nonlinear weighted averaging of nearby plumes, and Cartwright et al. (2021) develop an emulator that is capable of reconstructing LPDM footprints given a 'known' set of nearby footprints, using a convolutional variational autoencoder for dimensionality reduction



and a Gaussian process emulator for prediction. Though more computationally efficient than LPDMs alone, these methods still require running the LPDM a number of times for new predictions. An emulator that is capable of making footprint predictions without needing nearby simulator runs would allow substantial further efficiency gains.

Here, we present a machine learning emulator for backward running LPDM simulations, based purely on meteorological inputs. Our emulator outputs hourly footprints for a small (approx 350 by 230 km) region around an observation point. Once

trained it does not require any further 3D simulator runs for footprint prediction. The emulator can only be constructed for fixed measurement locations, and therefore it is not applicable for satellite retrievals. However, we present it as a proof-of-concept with a simple and interpretable design that can be built upon to be used for a wider range of measurement platforms. We train and evaluate the emulator by comparing to NAME for seven sites around the world, training with data from 2014 and 2015 and evaluating predictions for 2016 and 2020. In Sect. 2 we describe NAME, the training and testing data sets and the observation

locations. Sect. 3 outlines the machine learning model and its characteristics, and Sect. 4 and 5 demonstrate and evaluate the predictive capabilities of the emulators. In Sect. 6, we discuss the applicability of our methodology and potential avenues for further development.

## 2  Data and observations

### 2.1  Measurement locations

Our emulator is designed to be applied to the calculation of LPDM footprints for in situ measurement stations anywhere in the world. Here, we emulate the NAME model at seven locations. These locations were chosen to emulate a national network, so that inverse modelling of national emissions could be performed, and then two other locations in different meteorological regimes, to demonstrate versatility. The seven measurement locations are shown in Fig. 1. Five of these sensors, located in UK and Ireland, belong to the UK DECC (Deriving Emissions linked to Climate Change) Network (Stanley et al., 2018), and the

other two sensors belong to the AGAGE (Advanced Global Atmospheric Gases Experiment) Network (Prinn et al., 2018). The stations in the DECC network have been previously been used for evaluating UK's methane emissions using inverse modelling (Lunt et al., 2021), and the AGAGE stations, Trinidad Head and Gosan have been used in various inverse modelling studies (e.g. Ganesan et al., 2014). In Sect. 4.3, we detail further the characteristics of the $CH_4$ measurements and follow the method used by Western et al. (2021) to infer monthly UK emissions using the predicted footprints and compare the findings to those

of the NAME-produced footprints.

### 2.2  NAME model

The Met Office NAME model is used to produce the footprints to train and test emulators. Each footprint is producing by releasing 20,000 model particles from the inlet height, following them backwards in time for 30 days and tracking the time particles spent near the surface (defined as being below 40 meters above ground level). Output footprints have a resolution of

0.352°×0.234°(approximately 35x23km resolution in mid-latitudes).



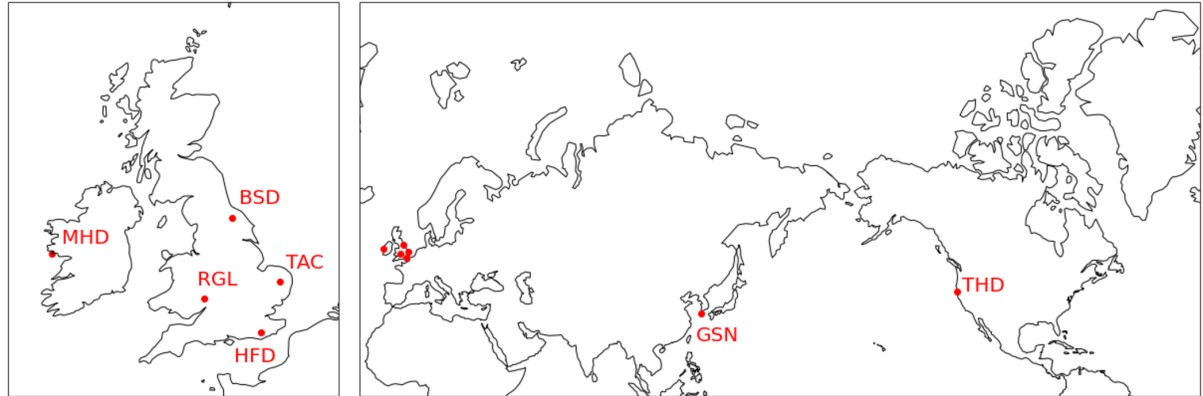

**Figure 1.** Measurement sites, in the UK and Ireland (left panel) and rest of the world (right panel). Sensors are located at Mace Head (MHD, Ireland, 53.326° N, 9.904° W, inlet is 10 meters above ground level), Ridge Hill (RGL, UK, 51.997° N, 2.540° W, 90 magl), Bilsdale (BSD, UK, 54.359° N, -.150° W, 250 magl), Heathfield (HFD, UK, 50.977° N, 0.230° E, 100 magl), Tacolneston (TAC, UK, 52.518° N, 1.139° E, 185 magl), Gosan (GSN, Korea, 33.292° N, 126.161° E, 10 magl) and Trinidad Head (THD, USA, 41.054° N, 124.151° W, 10 magl).

NAME was run using UKV meteorology (a UK-specific mesoscale meteorological analysis) over the UK, and global meteorology fields from the UK Met Office's Unified Model (UM, Cullen (1993)) everywhere else. UKV has a resolution of 1.5 km and 1h with 57 vertical levels up to 12 km, and UM meteorology has a resolution of 3h and 25 km up to July 2014, 17 km from then until July 2017 and 12 km after, with 59 vertical levels up to 29 km. The surface meteorology used as inputs is extracted
from the UKV meteorology for UK sites and the global UM meteorology for the other sites, and the vertical gradients used across all sites are also extracted from UM. Data from both models is interpolated linearly in time to increase resolution to hourly, and in space to the same resolution as the footprint, 0.352°×0.234°.

We use the computational domains to produce footprints with NAME used in previous studies (e.g. Lunt et al., 2021; Rigby et al., 2019). The computational domain for Europe (used for all UK and Ireland sites) covers 10.7–79.1° N and -97.9–39.4° E,
the USA domain (used for the Trinidad Head site) covers 8–59° N and 140–39.7° W and the East Asia domain (used for the Gosan site) covers -5.2–74.1° N and 54.5° E-168.2° W.

The data set consists of NAME footprints calculated every hour throughout 2014, 2015, 2016 and 2020 for each site. We divide this data set into a training set, comprising 2014 and 2015 for all sites, and a testing set used to evaluate the emulators, comprising of data immediately consecutive to the training data (2016) and five years after (2020). Each hourly footprint takes
about 10 minutes to be produced.





## 3    Emulator design

### 3.1    Formalisation

An LPDM $f$ produces a footprint $\boldsymbol{y}_t^{\phi}$ for receptor $\phi$ and time $t$, given the location and height of the receptor, the topography around it (both receptor location and topography summarised as $\phi$), and a time series of meteorological features $(\boldsymbol{x}_t)$, so that

$\boldsymbol{y}_t^{\phi} = f(\phi, \boldsymbol{x}_t)$.

An LPDM emulator $\hat{f}$ is a statistical approximation of $f$, built using simulator runs $f(\phi_m, \boldsymbol{x}_n)$. As this analysis comprises seven independent sites, we build instead site-specific emulators $\hat{f}^{\phi}(\boldsymbol{x}_t)$, each trained with data for a single location.

There are many potential approaches to inferring $\hat{f}$ using machine learning techniques: designing a model that can directly output 2D images, like neural networks; using a dimensionality reduction method to decompose $\boldsymbol{y}$ into a set of features and

coefficients and training a model to output coefficients given new inputs (e.g., Francom et al., 2019); or training a number of simple regressors where each outputs the value at a single location in $\boldsymbol{y}$ (e.g., Gunawardena et al., 2021). Each of these approaches has certain advantages and disadvantages. Models that are able to output 2D images directly involve deep learning, which can be difficult to design, train and interpret and are computationally expensive. Decomposing the data to reduce the problem's dimensionality is a common method in the Earth sciences, particularly using empirical orthogonal functions (EOFs).

However, Cartwright et al. (2021) demonstrate that EOFs are not able to retain the structural information of footprints as well as a deep learning alternative, which in turn requires additional complexity, including longer training and predicting times and rotating the footprints to reduce spatial variability. A gridcell-by-gridcell approach is simpler to design, train and interpret, but does not implicitly capture the spatial and temporal structure of the output.

### 3.2    Model design

As the work presented here is a proof-of-concept demonstrating that a few selected meteorology inputs can be used to produce footprints with reduced computational expense, we demonstrate the use of a gridcell-by-gridcell model. As each footprint $\boldsymbol{y}_t^{\phi}$ is a 2-dimensional grid, the value of the emulated footprint $\hat{\boldsymbol{y}}_t^{\phi}$ at each cell $(i,j)$ is predicted by an independent regression model $r_{i,j}^{\phi}$ using a subset of the meteorological inputs, such that $\hat{\boldsymbol{y}}_{t,i,j}^{\phi} = r_{i,j}^{\phi}(\boldsymbol{z}_{t,i,j})$ where $\boldsymbol{z}_{t,i,j} \subseteq \boldsymbol{x}_t$.

To reduce computational expense, we calculate the footprint only in a sub-domain of 10x10 cells centered around the

receptor, so that $i,j = 1,2,3...10$ with the receptor located at $(5,5)$. Therefore, each emulator is formed by 100 regressors.

We use Gradient Boosted Regression Trees (GBRTs) as regressors, as they are easy to build, can handle multi-collinearity in the inputs, are highly interpretable and have been used repeatedly in atmospheric science (e.g. Ivatt and Evans, 2020; Sayegh et al., 2016; Lucas et al., 2017). GBRTs are built of regression trees, a non-linear non-parametric predictive model also known as decision trees. Regression trees partition the input space recursively, once per node, making binary splits on the input data

(i.e. for sample $z$, is the value of feature $x_z$ bigger than value $k$?). The input space is therefore divided into regions, where each region corresponds to a terminal node or leaf. For any new data point, the value predicted will be a combination of the all the training samples in that leaf - for example, the mean if using mean squared error as a loss function, and the mode if using mean absolute error. Though useful, regression trees alone can be inaccurate and unstable. GBRTs use boosting to create





a more robust regressor: they are a sequence of regression trees, where each tree attempts to predict the errors of the sequence
before it (Friedman, 2001).

### 3.3   Model inputs

Each individual regressor takes as inputs meteorological variables ($z_{t,i,j}$) from grid cells at two sets of locations: at the cell
it is predicting $(i,j)$ and the eight adjacent cells, and at the measurement site $(5,5)$ and the eight adjacent cells. Therefore,
each regressor $r_{i,j}^{\phi}$ will have inputs from eighteen locations (which might overlap). This selection of meteorological inputs was
chosen because these two regions will dominate the footprint value at a given cell, with the meteorology at the measurement site
dictating the overall footprint direction and the local dynamics around a cell affecting the specific behaviour. Testing indicates
that this selection produces better predictions that providing the meteorological inputs at all locations, or at a fixed reduced set
of locations for all cells. The meteorological inputs used are the $x$ (West to East) and $y$ (South to North) wind vectors at 10 m
above ground level, planetary boundary layer height (PBLH), all taken both at the time of the footprint and six hours before,
as well as vertical gradients in temperature and $x$ and $y$ wind speed (between 150 m and 20 m). Other potential inputs, like sea
level pressure and absolute temperature were not used as they did not substantially increase the predictive power of the model.

An efficient emulator should train with as few samples as possible, while observing sufficient examples of the potential
meteorological configurations. We study the data needs of the model by training and evaluating emulators using all the training
data set, a half, a quarter and a sixth of the data set (hourly, 2-hourly, 4-hourly and 6-hourly footprints respectively), where
the hourly data set has 17520 samples. We find that there is no difference in emulation quality between training with hourly
and 2-hourly data, that the 4-hourly data produces noisier footprints and that the 6-hourly data has little prediction power. We
therefore choose to train the model with 2-hourly data, needing around 8700 footprints to train at each site.

### 3.4   Evaluation metrics

This section outlines the evaluation metrics used to assess the quality of the footprints predicted by the emulators, and where
each of them is applied.

– R-squared score ($R^2$): Also known as coefficient of determination, $R^2$ represents the proportion of the variance in the
dependent variable that is explained by the independent variables (Chicco et al., 2021). It is defined as $R^2(a,\hat{a}) =
1 - \sum_{i=1}^{m}(\hat{a}_i - a_i)^2 / \sum_{i=1}^{m}(\bar{a} - a_i)^2$, where $\hat{a}$ are the predicted values and $a$ the real values. It can range between -$\infty$
and 1, where 1 represents perfect predictions, 0 means none of the variance is explained (the model predicts the mean of
the data at all points), and below 0 means an arbitrarily worse model.

– Mean Bias Error (MBE): MBE measures any systematic errors in the predictions, and is defined as $MBE(a,\hat{a}) =
\frac{1}{m}\sum_{i=1}^{m}(\hat{a}_i - a_i)$. A positive MBE indicates that the model tends to over predict the output, and a negative MBE the
opposite.





   – Mean Absolute Error (MAE) and Normalised MAE (NMAE): NMAE is the MAE normalised by the mean of the true
data, so the metric can be comparable across data sets of different scales. It is defined as $NMAE(a, \hat{a}) = \frac{1}{m\bar{a}} \sum_{i=1}^{m} |a_i - \hat{a}_i|$.
      Lower values represent better predictions.

   – Accuracy (AC): Accuracy is used to calculate the spatial agreement of the footprints, measuring which percentage of
      cells is correctly emulated to be above or below a threshold $b$. A binary mask is created where the values of the footprint
      surpass $b$, such that

$$\boldsymbol{Y}_{t,i,j}^{\phi} = \begin{cases} 1 & \text{if } \boldsymbol{y}_{t,i,j}^{\phi} > b \\ 0 & \text{otherwise} \end{cases} \tag{1}$$

      and similarly $\hat{\boldsymbol{Y}}_{t,i,j}^{\phi}$ using $\hat{\boldsymbol{y}}_{t,i,j}^{\phi}$. The accuracy of an emulated footprint $\hat{\boldsymbol{y}}_t^{\phi}$ is therefore calculated with $AC = 100\% \times \frac{\left|\boldsymbol{Y}_t^{\phi} = = \hat{\boldsymbol{Y}}_t^{\phi}\right|}{\left|\boldsymbol{Y}_t^{\phi}\right|}$. A higher accuracy indicates better spatial agreement above threshold $b$ of the emulated and real footprint.

The emulated footprints are evaluated in three different ways: 1) footprint-to-footprint comparison, 2) convolving the foot-
prints with a surface emissions inventory to obtain the above-baseline mole fraction, and 3) conducting a flux inversion to
estimate UK methane emissions. We do footprint-to-footprint comparison using accuracy to measure the spatial agreement
of the footprints and NMAE. We use R-squared score, NMAE and MBE to evaluate the predicted mole fractions. The UK
methane monthly emissions from the emulated footprints and the real footprints are compared using MAE.

## 3.5 Training the model

We tune the hyper-parameters for each of the emulators optimising for R-squared score between the true footprint value at a
particular cell and its prediction. This metric is chosen as opposed to other common metrics like Mean Square Error (MSE) or
Mean Absolute Error (MAE) because the range of values in each cell varies with its position with respect to the release point.
As R-squared score does not depend on the distribution of the ground truth, it is easily interpretable across regressors.

   We use 3-fold validation for ten random regressors in each site, finding that the chosen parameters barely change across cells
and sites, and therefore we select the hyper-parameters to be equal throughout the emulators. As expected, we find that deeper
trees perform better as they are able to capture better high-order interactions than shallow trees (Friedman, 2001). In this case,
we require a maximum depth of at least 50 nodes. We find that at least 150 trees in each GBRT with a learning rate of 0.1 is
preferred, with the first trees having most of the predictive power and the bulk of the trees providing small improvements to the
R-squared score. We also find that the absolute error is a better loss function than mean-squared error. This is likely because
the data is approximately exponential in distribution, with most of the values for each regressor being zero or near-zero except
a few spikes or outliers. MAE is a more appropriate metric for the Laplacian-like errors often produced by exponentially
distributed variables (Hodson, 2022). Moreover, as shown often in literature, adding randomness to the GBRT also increases
the score (Friedman, 2002). We find that training each tree with randomly selected $\sqrt{n}$ features, where $n$ is the total number of
features, increases the training score significantly as well as reducing the computational expense. However, there is no benefit
to using data subsampling.





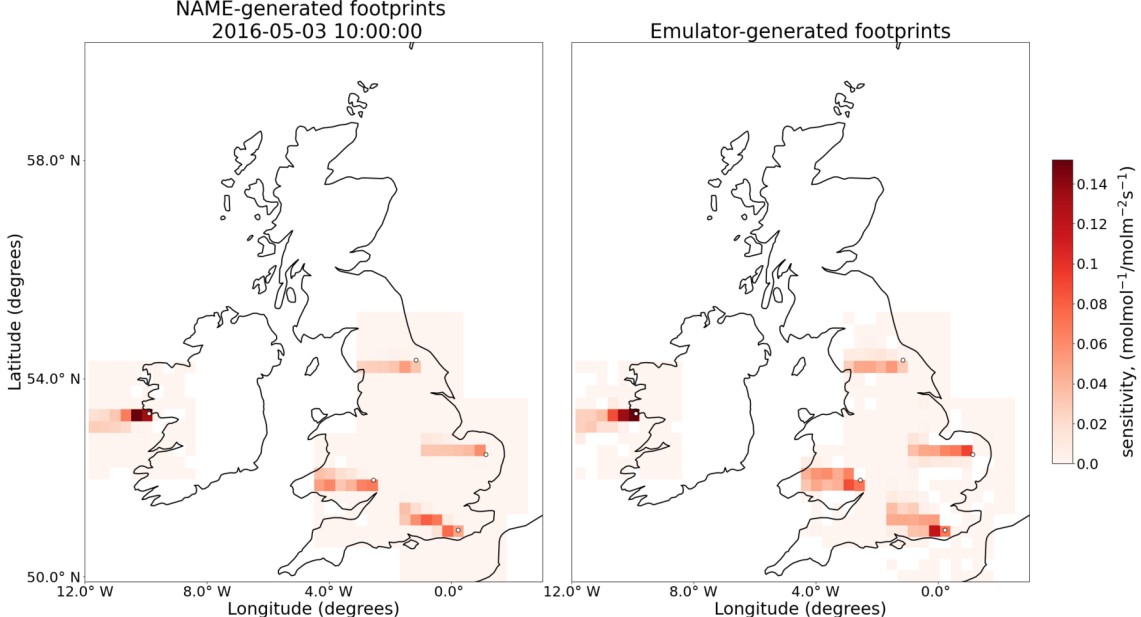

**Figure 2.** NAME-generated footprints (left) and emulator-generated footprints (right) for the same date (3 May 2016, 10:00:00) and the five sites in UK and Ireland (Mace Head, Bilsdale, Ridge Hill, Heathfield and Tacolneston, marked with a white dot). In the cells where domains for different emulators overlap, the sensitivity represents the maximum value across emulators for that cell.

## 4 Results and Discussion

The emulator is trained for the seven locations shown in Fig. 1, using footprints every two hours from 2014 and 2015. In this section, we evaluate the hourly footprints these emulators produce for 2016 and 2020 (i.e., there is no overlap between the meteorology used to train the emulators and that used to test them). Figure 2 shows an example of five emulated footprints for the DECC network sites at a particular date. The predictions are evaluated in three different ways: footprint-to-footprint comparison, predicted mole fraction evaluation, and UK inversion results.

### 4.1 Footprint-to-footprint comparison

We compute the Normalised Mean Absolute Error (NMAE) for every predicted footprint, averaging the error throughout the cells. We find that across all footprints and sites, the NMAE is 0.689 for 2016 and 0.701 for 2020. We also compute the accuracy of the footprints, which estimates which percentage of the cells is correctly emulated to be above or below a footprint value threshold $b$ (see Sect. 3.4). We find that across sites, the emulated footprints have an accuracy of 67.3% and 64% for 2016 and 2020 respectively with $b = 0$, and of 88.1% and 87.8% respectively for $b = 0.01$. Figure 3 shows the NMAE and accuracy for each of the sites.





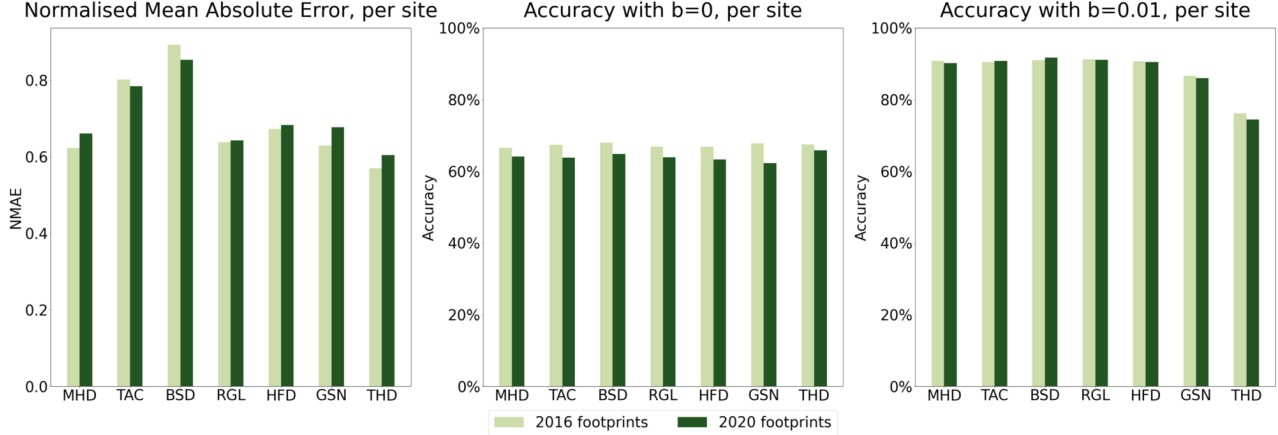

**Figure 3.** Evaluation of emulators with footprint-to-footprint comparison, using metrics NMAE and accuracy with $b = 0$ (all footprint values) and $b = 0.01$ (high values), shown per site and year.

## 4.2 Predicted mole fractions

The LPDM footprint can be convolved with a map of gridded emissions to provide the expected above-baseline mole fraction

for that measurement location and time. This is calculated doing element-wise multiplication (Hadamard product) of the two grids, and summing over the area. Here, we generate pseudo time series of atmospheric methane, but our evaluation could readily have used any other species. When applied to the emulated footprints, this produces an emulated time series of expected $CH_4$ concentration in the area that can be compared to the NAME-generated $CH_4$. Figure 4 shows two month-long examples, March 2016 and October 2020, of the time series obtained from the emulated and NAME-generated footprints.

We use EDGARv6.0 (Crippa et al., 2021) for 2016 as the gridded emissions for both 2016 and 2020, as the 2020 data set has not been released yet. EDGARv6.0 represents the mean yearly emissions on a $0.1° \times 0.1°$ resolution, which is regridded using an area-weighted scheme to the same resolution as the footprints.

We calculate the NMAE, R-squared score and MBE for every time series. We find that across all sites, the NMAE is 0.308 for 2016 and 0.308 for 2020, the R-squared score is 0.694 and 0.697 respectively, and the MBE is -0.0125 micro mol mol$^{-1}$

and -0.0043 mol mol$^{-1}$ respectively. Figure 5 shows these metrics for each of the sites.

Although the MBE for the emulator is small, Fig. 4 shows that the highest mole fractions are often not well predicted. We show how the bias changes across values by dividing the yearly real data into 10-quantiles (10 equal-sized, ordered subsets) and taking the MBE of each. Figure 6 shows that the bias is small for lower molar fraction values, but that the model tends to highly under-predict the higher range, with a similar behaviour across sites.





**Figure 4.** Mole fractions from NAME-generated footprints and predicted footprints, for March 2016 (left column) and October 2020 (right column) around each measurement site.



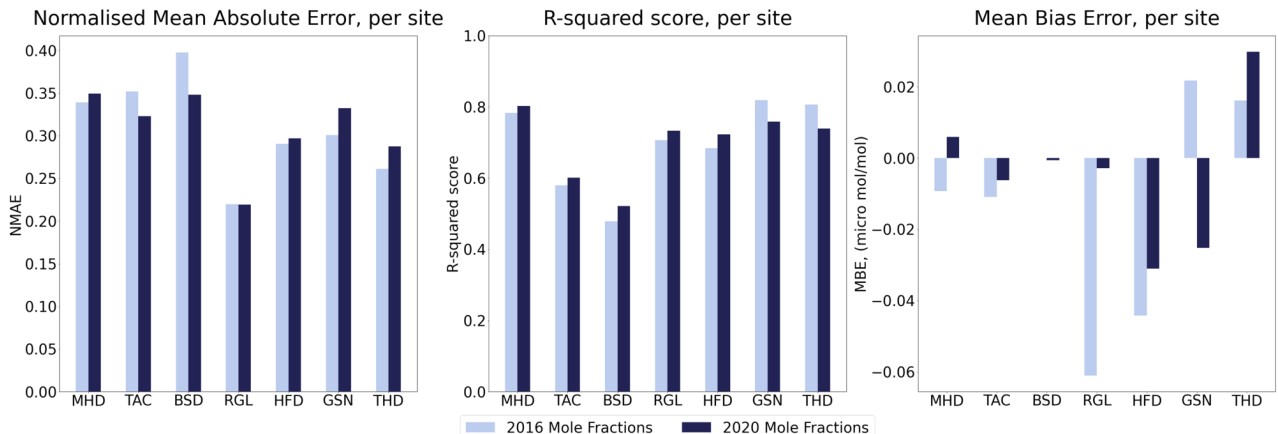

**Figure 5.** Evaluation of emulators with mole fraction comparison, using metrics NMAE, R-squared score and MBE, shown per site and year.

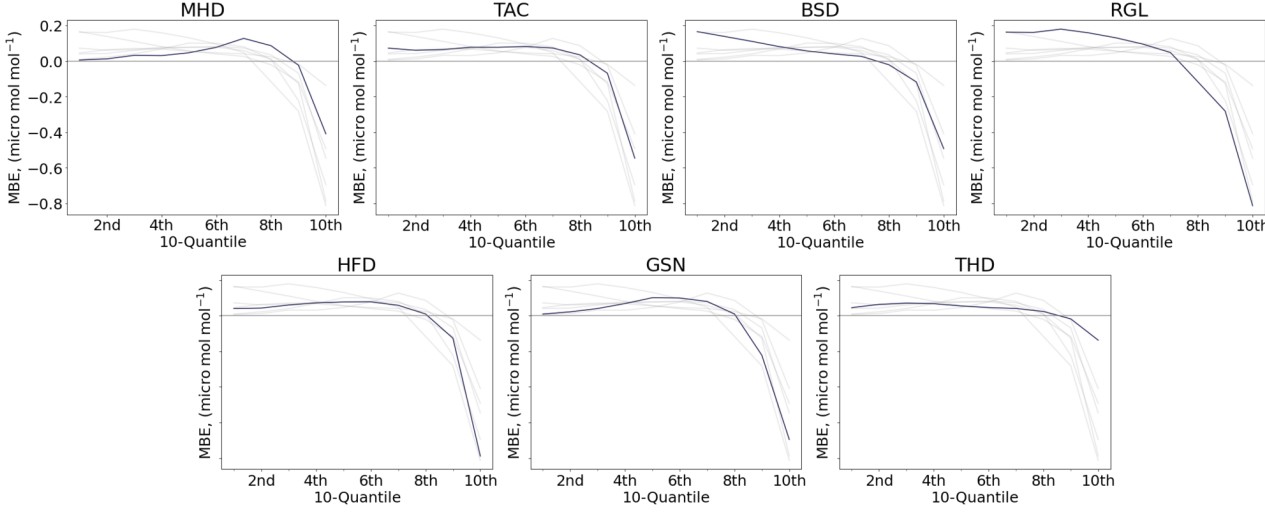

**Figure 6.** Mean Bias Error in the above baseline mole fraction predictions for each of the 10-quantiles of the true data, across sites for 2016. An MBE above zero indicates overprediction, and below zero indicates underprediction.

## 4.3 UK emissions inversion

To evaluate the performance of the emulator in a common application we carry out a UK methane flux inversion that has recently been performed in Lunt et al. (2021, 2016) and Western et al. (2020). We follow a Hierarchical Bayesian Markov chain Monte Carlo (MCMC) method and use input parameters described by Western et al. (2021) to estimate monthly UK methane emissions for 2016 using the predicted footprints. We use the DECC network sensors, which have measured CH$_4$ continuously for the period analysed here: Mace Head, Ridge Hill, Bilsdale, Heathfield and Tacolneston (Fig. 1). Details on





the prior and instruments used for measurements can be found in Lunt et al. (2021), but note that a slightly different inversion method is used in that paper.

As our emulator is predicting footprints in a small domain around the sensor and the inversion requires a bigger domain, we produce an estimate of the total footprint by using the NAME-calculated footprint in the rest of the domain not within our
emulated region. Therefore, while our emulator calculates the most important part of the footprint (i.e., the part with the highest values), it should be noted that there will be some influence of the "true" footprints on the final results for this comparison.

Using the NAME-generated footprints, we find a UK mean for 2016 of 2.03 (1.90-2.16) Tg yr$^{-1}$ (uncertainty represents 95% High Density Interval), consistent with top-down estimates in Manning et al. (2020) and Lunt et al. (2021) and with the 2016 inventory (Brown et al., 2020). Using the full-domain emulated footprints, the UK mean for 2016 is 2.15 (2.02-2.30) Tg yr$^{-1}$,
5.9% higher than inferred with the real footprints. The monthly emission rates can be seen in Fig. 7, with a mean monthly difference between the real and predicted inversion of 0.130 Tg, or 6.32%. This increase in inferred emissions, compared to the inversion using the real footprints, is consistent with the emulator generally under-estimating the highest mole fractions.

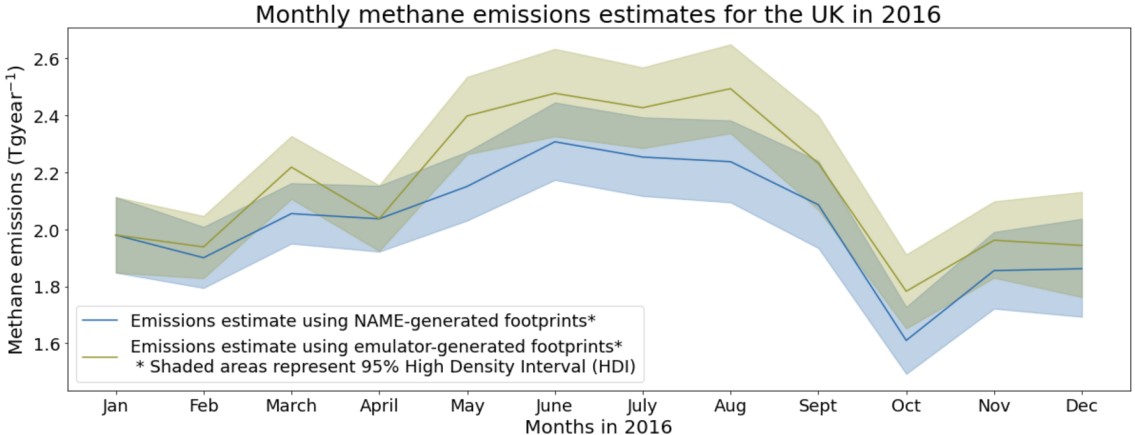

**Figure 7.** UK monthly methane emission estimates using the NAME-generated footprints for the five sites in the DECC network (UK and Ireland sites) and the emulator-generated footprints (with NAME-generated values outside of emulated region) for the same sites.

## 5 Feature evaluation

The design of the emulator, with one regressor per cell, means we can evaluate which input variables are more relevant at
each cell and therefore understand the spatial distribution of the feature importances, and check if they are physically coherent. Tree-based models like GBRTs are highly interpretable as they can rank the inputs in terms of how much they contribute to building the trees. However, when working with multi-collinearity in the inputs these feature importances are not reliable, as similar information is present across correlated features.





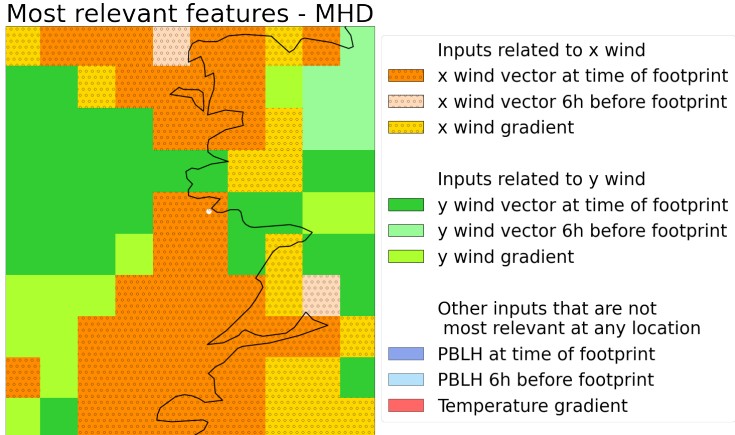

**Figure 8.** Most important block of meteorological features at each cell for Mace Head, as calculated with the block permutation importance.

Another common way to rank the features in a model is by calculating the feature permutation importance. In this approach, the values of one or more inputs in a data set are shuffled, effectively adding noise to that feature, and the prediction error of the new data set is compared to the prediction error of the original data set, called baseline error (Molnar, 2022). If used on one feature at the time on correlated inputs, this approach has similar issues to the GBRTs' importance. However, it can be run on multiple features at the time, meaning that we can calculate the importance of blocks of correlated features.

To calculate the feature importances across the domain, we divide the input data for each regressor into nine blocks, one per meteorological input across all locations: $x$ wind (West-East), $y$ wind (South-North) and PBLH at the time of the footprint, the same three inputs six hours before the sampling time, the $x$ and $y$ wind vertical gradients and the temperature gradient. We calculate the baseline NMAE for each regressor using data for 2016, and then calculate the NMAE when shuffling each of the blocks. The difference between the two, the added error, is a proxy for feature importance - the more a regressor relies on a feature, the higher errors it will produce when that feature is noisy. As an example, Fig. 8 shows the most important feature at each cell for Mace Head. We propose that the distribution of importances can be interpreted in physical terms as follows: cells on the S-N axis are more affected by the E-W wind, as low E-W winds would mean higher concentrations in said axis and vice-versa, and similarly for the E-W axis and S-N winds. Moreover, winds six hours before the footprint become increasingly relevant towards the edges of the domain, consistent with the dispersion running backwards in time.

## 6  Conclusions

We have presented a proof-of-concept emulator for LPDMs, which can efficiently produce footprints using only meteorological inputs, and we demonstrated its performance on seven measurement stations around the world. The emulator offers a considerable speed-up with respect to both normal LPDM runs and interpolation-based methods, because once trained, it does not require further LPDM runs. The emulator can produce footprints that resemble those generated by NAME, with high





correlations and low bias for the predicted above-baseline mole fraction at the seven sites investigated here. An inversion of
UK methane fluxes performed using our emulated footprints was not statistically different to an inversion using the LPDM
footprints. As there is no decrease in performance between 2016 and 2020, the emulator appears to have inference capabilities
for at least five years after the training data, making it a long-term tool that does not require retraining often. Moreover, we use
meteorology at different resolutions in different sites (high-resolution, national UKV and coarse resolution, global UM) but we
see no differences in scores across several metrics, meaning the model can be trained and used with different input resolutions.
Although not validated in this work, it is likely that performance will be similar when training with different LPDMs, like
STILT or FLEXPART.

There are limitations in our emulator that will need to be overcome before it could be used to replace LPDM model evalu-
ations in applications such as inverse modelling. Our emulator predicts only a small domain around the receptor, which is not
big enough for most national-scale inversions. The domain size could be increased by using extra regressors. However, as the
training time increases linearly with the number of regressors, strategies to keep training times feasible should be considered.
This could include coarsening the grid towards the edges of the domain or parallel training. Further work would be required
to select the most appropriate combination of input data for the added regressors, including, for example, meteorology further
back in time. The design of the emulator, chosen due to its simplicity to set up and train, could also be improved by making
the regressors dependent, either across time (a regressor's inputs include data from the previous footprint) and/or across space
(a regressor's inputs include predictions from nearby regressors).

As well as the known design limitations, the performance of the emulator highlights opportunities for improvement. The
high-value bias, present in all sites, could be reduced with approaches such as bias reduction methods. Identifying the meteo-
rological conditions in which the model performs more poorly would also be useful, in particular, to relate them to conditions
usually filtered out in inversions or in which the LPDM is also considered less reliable. For example, low wind conditions,
which usually cause high local influence, could coincide with the badly-predicted events. Creating a training data set with a
more balanced distribution of meteorological conditions may also help reduce the differences in performance across different
situations.

The difference in prediction quality across sites could also provide insights into potential areas of improvement for the
model. We find that there are no noticeable differences in accuracy with $b = 0$ across sites, meaning that the spatial distribution
of footprints is captured similarly. The improvement in accuracy between thresholds $b = 0$ and $b = 0.01$ likely indicates that
the emulators are better at capturing the main shape of the footprint, composed of higher values (i.e. $b = 0.01$), but that the
background (captured with $b = 0$) tend to be less well predicted - for example, in Fig. 2 it can be seen that the model confuses
very small values with zero. We find however a difference in performance across sites when when evaluating the values
predicted with NMAE. We find that the receptors close to ground level (MHD, GSN and THD are at 10 meters above ground
level) are significantly better predicted that those with higher inlets (TAC is 185 magl and BSD is 250 magl), both when doing
footprint-to-footprint comparison and when evaluating the predicted mole fractions. As most of the inputs are provided at 10
magl, when PBLH is low this meteorology may not be representative of the state of the atmosphere around the taller sensors
and therefore lead to higher errors.



To more fully exploit the possibilities of machine learning, it would be desirable to generalise the emulator to any location. For the emulators built here, the effect of the surface surrounding each site is implicitly captured by using site-specific training data. For the emulator to be applied at an arbitrary location, it should have knowledge of the effect of topography and other surface characteristics on dispersion. A well-designed emulator that is trained with data for some locations should therefore be able to produce footprints for similar, unseen locations. Use of additional variables (e.g., vertical wind speed, etc.) and designing the model to read and exploit 2D, 3D or 4D meteorological fields may improve prediction accuracy. Ideally, more advanced models should also estimate an uncertainty in the predictions, either directly through the model or by choosing a probabilistic method that can be used to build ensembles.

*Code and data availability.* Code used to train and evaluate the models is available as a free access repository at DOI 10.5281/zenodo.7254667 (Fillola, 2022b). Sample data to accompany the code, including the trained emulator for Mace Head and inputs/outputs to test it, can be found at DOI 10.5281/zenodo.7254330 (Fillola, 2022a). The NAME III v7.2 transport model is available from the UK Met Office under licence by contacting enquiries@metoffice.gov.uk. The meteorological data used in this work from the UK Met Office operational NWP (Numerical Weather Prediction) Unified Model (UM) are available from the UK Centre for Environmental Data Analysis at https://data.ceda.ac.uk/badc/ukmo-nwp/data, at the UKV (Met Office (2013), a; Met Office (2016), a) and global (Met Office (2013), b; Met Office (2016), b) resolutions. The software used for the inversion can be found at DOI 10.5281/zenodo.6834888 (Rigby et al., 2022). Measurements of methane for the Mace Head station are available at http://agage.mit.edu/data (Prinn et al.) and measurements of methane from the UK DECC network sites Tacolneston, Ridge Hill, Heathfield and Bilsdale are available at https://catalogue.ceda.ac.uk/uuid/f5b38d1654d84b03ba79060746541e4f (O'Doherty et al., 2020).

*Author contributions.* EF carried out the research and wrote the paper, with input from all co-authors. The research was designed by EF, MR and RS. AM produced the footprint data and SO provided the atmospheric measurements.

*Competing interests.* We declare no competing interests.

*Acknowledgements.* EF was funded through a Google PhD Fellowship. MR was funded through the Natural Environment Research council Constructing a Digital Environment OpenGHG project (NE/V002996/1). RS was funded by the UKRI Turing AI Fellowship EP/V024817/1. Measurements from Mace Head were funded by the Advanced Global Atmospheric Gases Experiment (NASA grant NNX16AC98G) and measurements from the UK DECC network by the UK Department of Business, Energy & Industrial Strategy through contract (1537/06/2018) to the University of Bristol. Since 2017, measurements at Heathfield have been maintained by the National Physical Laboratory mainly under funding from the National Measurement System. This work was carried out using the computational facilities of the Advanced Computing Research Centre, University of Bristol - http://www.bristol.ac.uk/acrc/.



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
