# Peer review of "A machine learning emulator for Lagrangian particle dispersion model footprints: a case study using NAME"

_EGUsphere, 2022_

## Author Response (AR1)

We thank the reviewers for their thoughtful comments and have addressed them below. The responses to the comments are shown in blue.

**Reviewer 1**

The authors motivate the study by citing the need for footprints for atmospheric inversions. However, the emulated footprints do not cover enough of the spatial domain to perform an inversion for the UK and to perform the inversions they borrow the un-emulated portions from the real footprints. The authors acknowledge this limitation and point out that they emulate the most significant features of the footprint. However, it is still unsatisfying that they are unable to use the emulator for the primary reason that it was developed. I think some of these concerns would be alleviated if they could quantify the error associated with using an incomplete footprint for their inversion.

The reviewer is correct that although one ultimate goal for our LPDM emulator is to accelerate the inversion, the current proof-of-concept model cannot be used for this purpose, primarily because only a subset of the full domain is emulated. We certainly plan to address this issue in the next iteration of our algorithm but feel that the innovations presented in this paper are sufficiently novel and timely to warrant publication as an initial step. Furthermore, there are useful outcomes that can be derived directly from the results of this study, such as efficiently characterising the influence of nearby emissions on trace gas measurement sites.

The purpose of the UK methane inversion that we present was to provide a metric with which to judge the emulator performance, in addition to the other metrics that are presented (e.g., normalised mean absolute error of the footprint, the R-squared score of the mole fraction simulations). To do this,  we had to use the un-emulated portions from the real footprints, which of course compromises this test somewhat, as the reviewer points out.

We do not believe that it is possible to quantify the error associated with using the incomplete footprint in our inversion. Instead, to address the reviewer's comment, we now add a sensitivity test that aims to show the dependence of the inversion result on the correctness with which the footprint is simulated within and outside the emulated region. To do this, we coarsen the emulated footprints (composed of the emulated region and the NAME-generated data in the rest of the domain) by dividing the image into independent windows of size *FxF* cells where *F* is the coarsening factor, and in each window we replace the value of all cells with the mean. This emulates running the LPDM at lower resolutions.

Figure R1a shows the results of running the inversion with the emulated footprints coarsened to different factors throughout the entire domain. Figure R1b shows the inversion estimates with the same coarsened footprints, but with the emulated region at full resolution. For efficiency, we obtain the emission estimates using the maximum a posteriori (MAP) probability rather than running the full Markov chain Monte Carlo (MCMC) sampling, but we show that the two are very similar compared to the a posteriori uncertainty by generating the MAP estimates for the full resolution footprints (shown as dotted lines in figure R1) and comparing them to the MCMC estimates.

[Figure]

**Figure R1.** Monthly methane emission estimates (as shown in Fig. 7 in the manuscript) compared to emission estimates from coarsened footprints, without the full-resolution emulated area (a) and with it (b), calculated with Maximum A Posteriori (MAP). In both graphs, the MAP emissions estimates for the NAME-generated and the emulator-generated footprints are shown as dotted lines, to demonstrate that the MAP approach and the MCMC approach are equivalent. The inversion is performed for the emulated footprints coarsened to five different coarsening factors (where coarsening means dividing the image into independent windows of size *FxF* cells where *F* is the coarsening factor, and in each window replacing the value of all cells with the mean).

The mean percentage absolute error between the emissions inferred using full-resolution emulator-generated footprints and those using the coarsened footprint is around 10% for a coarsening factor of 5, and over 40% for a coarsening factor of 30. In comparison, the error for the coarsened footprint outside of the local area present errors of under 5% for all coarsening factors. This indicates that the inversion is highly sensitive to a loss of fidelity in the simulated footprints within our emulated region, but substantially less sensitive outside of this region. Therefore, we propose that our test, whilst necessarily incomplete for this proof-of-concept, does provide an indication that our inversion results should be relatively insensitive to substantial uncertainties in footprint magnitude outside of the emulated regions. Furthermore, we note that this test suggests that a low-resolution emulator further away from the measurement site may be a sensible next step for deriving an emulator for the full domain.

We have added the following in line 252 in the manuscript, and a description of the sensitivity test, the results and figure R1 in Supplement B.

Line 252: "We conduct a sensitivity test in Supplement B which demonstrates that the inversion is highly sensitive to the emulated area, but less outside of this region."

Supplement B:

"We conduct a sensitivity test to demonstrate the importance of the emulated area (the local region around the measurement point) in comparison to the rest of the domain by calculating emission estimates with coarsened footprints, simulating lower resolution runs of the LPDM. We coarsen the emulated footprints (which consist of the local emulated area of size 10x10 and the NAME-generated data in the rest of the domain) by dividing the image into independent windows of size FxF cells where F is the coarsening factor, and in each window we replace the value of all cells with the mean.

Figure B1a shows the results of running the inversion with emulated footprints coarsened to different factors throughout the entire domain. Figure B1b shows the inversion estimates with the coarsened emulated footprints, but the local area preserved at full resolution. For efficiency, we obtain the emission estimates using the maximum a posteriori (MAP) probability rather than running the full Markov chain Monte Carlo (MCMC) sampling, but we show that the two are very similar compared to the a posteriori uncertainty by generating the MAP estimates for the full resolution footprints (shown as dotted lines in figure B1) and comparing them to the MCMC estimates.

The mean percentage absolute error between the emissions estimate inferred using full-resolution emulator-generated footprints and the estimate using the coarsened footprint is around 10% for a coarsening factor of 5, and over 40% for a coarsening factor of 30. In comparison, the error for the coarsened footprint with the emulated area at full resolution present errors of under 5% for all coarsening factors. This indicates that the inversion is highly sensitive to a loss of fidelity in the footprints within our emulated region, but substantially less sensitive outside of this region. We propose this test provides an indication that our inversion results should be relatively insensitive to substantial uncertainties in footprint magnitude outside of the emulated regions."

**Specific comments**

Lines 24-28: Is there a metric that the authors can provide that provides context on the number of fixed-site observations? As the authors acknowledge, the emulator presented here cannot emulate footprints for satellite measurements and thus the number of satellite observations is not particularly relevant.

It is somewhat difficult to determine the number of operational measurement stations worldwide. The Global Atmosphere Watch Station Information System shows around 150 land-based fixed sites at the time of writing. However, most of these are low-frequency flask-sampling locations, rather than "high frequency" measurement stations. To provide some rough context that can be contrasted with satellite data, we have added to line 23:

"(tens of sites globally that together collect ~thousands of observations per month)"

Line 75: The authors state that the emulator is designed to be applied to calculate footprints anywhere in the world

Since we cannot show conclusively that we can apply this method everywhere in the world, we have deleted this part of the sentence.

Line 89: Why was 40 m chosen as the near-surface altitude? Could this choice help explain why the sites with higher inlets typically had poorer performance?

The altitude at which particles could be considered to interact with the surface was chosen somewhat arbitrarily based on several previous studies by the modelling team. It is based on several factors including the need to compile sufficient statistics on particle interactions with the ground whilst maintaining the influence of non-uniform distributions within the planetary boundary layer, the possibility of intercepting elevated sources (e.g., smokestacks), and the typical height of inlets in tall-tower monitoring sites. Since this setup has been used extensively, and is not important for our emulator evaluation, we do not think it would be worth discussing further in the text. However, we do now include citations where it has been used before in line 90. (Lunt et al., 2021; Lunt et al., 2016).

We believe that the lower performance for higher inlets is likely to be caused by the height of the selected meteorological inputs, 10 magl, which could be not representative of the meteorology at higher altitudes, rather than the near-surface altitude selected for the NAME runs.

Line 90: I know this is mentioned elsewhere, but it would be helpful to clarify here that the emulated footprints are 10 x 10 cells around the receptor site.

We have clarified this by adding "and cover a domain of 10x10 cells around the measurement site." to line 92.

Lines 92-94: The spatial resolution for UKV and UM are quite different—did this necessitate any differences in how the LPDM was run?

Yes, the model timestep is shorter for the UKV runs. Because the development of our emulator is not dependent on these modelling details (although of course the footprints themselves may be influenced), we do not feel that further discussion is warranted here.

Line 150: Are the vertical gradients also taken at time of footprint and 6 hours before?

No, the vertical gradients are only taken at the time of the footprint. We have clarified this in the manuscript by adding "taken only at the time of the footprint" in line 152.

Line 155: How long did it take to train the emulator?

The emulator for each site takes around 90 minutes to train on a 24-core CPU, this has been added to the manuscript in line 162: "The predictor for each cell takes under one minute to train in a 24-core CPU, meaning that the emulator described here for a 10x10 cells domain takes around 90 minutes"

Line 172: Is the threshold value b an absolute or relative value?

b is an absolute threshold. This has been clarified in the manuscript in line 182.

**Technical comments**

Line 8: "gradient-boosted regression trees" does not need to be abbreviated because it is not used later in the abstract.

Corrected.

Line 57: "the the full footprint" should be "the full footprint".

Corrected.

Line 257: Oddly worded sentence.

This sentence has been reworded for clarity to "This approach has similar issues to the GBRTs' importance if it is used on one feature at the time on correlated inputs." in line 269.

Line 305: "when PBLH" should be "when the PBLH".

Corrected.

**Reviewer 2**

**Baseline for comparison -** The GBRT method developed here performs well when compared to the full NAME model. Despite this good performance, GBRTs are still complex models that are nontrivial to develop and implement. A simple baseline like a linear model (similar to Barnes et al., 2020; Silva et al., 2022; Watson-Parris et al., 2022) would allow for readers to better understand just how good the GBRT predictions actually are.

To compare the GBRTs with a linear model, we use the same training and testing inputs and cell-by-cell approach but replace the GBRT with a ridge regression model. Ridge regression is a type of regularised linear regression, where an L2 penalty term is added to the loss function to shrink the regression coefficients, aiming to reduce the impact of collinearity in the data. The amount of shrinkage is controlled by a parameter often called "lambda".

The ridge regression model is tuned in a similar fashion to the GBRT model (see section 3.5), and we find that an lambda value of 1 is most fitting. We evaluate the footprints output by the linear emulation in the same way as the GBRT-generated ones. The results for the whole dataset are summarised in the table below, and figure R2 shows the evaluation disaggregated by site. These results demonstrate, in particular through the NMAE, that the GBRT model has far higher predictive skills than a linear model.

| Metric | GBRT model | | Ridge regression model | |
|---|---|---|---|---|
| | 2016 | 2020 | 2016 | 2020 |
| NMAE (footprints) | 0.689 | 0.701 | 1.12 | 1.15 |
| Accuracy with b=0 | 67.3% | 64.0% | 66.3% | 64.4% |
| Accuracy with b=0.01 | 88.1% | 87.8% | 75.7% | 75.1% |
| NMAE (mole fraction) | 0.308 | 0.308 | 0.514 | 0.655 |
| R-squared score | 0.694 | 0.697 | 0.451 | 0.283 |
| Mean Bias Error (micro mol/mol) | -0.0125 | -0.0043 | 0.0978 | 0.204 |

**Table R1.** Average metric results for 2016 and 2020 across all seven sites tested, for the GBRT model (the emulator described in the manuscript) and a ridge regression model (a baseline linear model).

**a)**

[Figure]

**b)**

[Figure]

**Figure R2.** Evaluation of emulators, per site and per year, for the GBRT model and a baseline ridge linear model. a) shows footprint-to-footprint comparison, using metrics NMAE and accuracy with b=0 (all footprint values) and b=0.01 (high values), and b) shows mole fraction comparison, using metrics NMAE, R-squared score and MBE.

The following text has been added to the manuscript in line 214:

"We also train a linear baseline model with the same data and structure, to evaluate the benefit of using GBRTs compared to a simpler model. More details and the full results are shown in Supplement A."

And the following is added to Supplement A, as well as table R1 and figure R2:

"We compare the performance of the emulator against a baseline ridge linear model, using the same training and testing data and the same cell-by-cell approach. Ridge regression is a type of regularised linear regression, where an L2 penalty term is added to the loss function to shrink the regression coefficients, aiming to reduce the impact of collinearity in the data. The amount of shrinkage is controlled by a parameter lambda. See Hastie et al. (2001, Section 3.4.1) for more on ridge regression.

The ridge regression model is tuned in a similar fashion to the GBRT model (see section 3.5), and we find that a lambda value of 1 is most fitting. We evaluate the footprints output by the linear emulation in the same way as the GBRT-generated ones. The results for the whole dataset are summarised in table A1, and figure A1a and A1b shows the evaluation disaggregated by site (the GBRT results are shown as well in figures 3 and 5 respectively). These results demonstrate, in particular through the NMAE, that the GBRT model has far higher predictive skills than a linear model."

**Training Time -** The GBRT the authors train are significantly faster than running the NAME model. However, the NAME model needed to be run to generate this training data. It would be useful context to know at what point the tradeoff between costly training data generation and cheap emulator execution comes out in favor of the emulator.

The reviewer makes a good point, as the NAME footprints are computationally expensive to generate. Each emulator was trained for the years 2014 and 2015, using approximately 8700 footprints (one footprint every two hours). Each footprint takes approximately 10 minutes to produce, so around 60 days of CPU time are required to produce the training dataset. The emulator can produce a footprint in 10ms (1.5 minutes for one year of hourly footprints) and takes 90 minutes to train. Therefore, if

more than approximately 8700 footprints are required for a particular site (around 1 year of hourly averages), it becomes more efficient to train the emulator than to perform further 3D model simulations. This has been developed in the paper in line 161:

"[…] needing around 8700 footprints to train at each site. As it takes around 10 minutes to produce one footprint with NAME, the training dataset for each site takes around 60 days of CPU time. The predictor for each cell takes under one minute to train in a 24-core CPU, meaning that the emulator described here for a 10x10 cells domain takes around 90 minutes and once trained, it can produce a footprint in 10ms (1.5 minutes for one year of hourly footprints). Therefore, if more than approximately 8700 footprints are needed for a particular site (around 1 year of hourly averages), it becomes more efficient to train the emulator than to perform further 3D model simulations (notwithstanding uncertainties as discussed in Section 4)."

**Differences between 2016 and 2020 -** A few of the figures (e.g., Fig 3) show results from 2016 and 2020 side by side. Are the differences between the cases significant, or should they be interpreted as generally providing the same information?

The differences between the 2016 and 2020 results are not significant, and therefore we understand that the emulator has inference capabilities of at least 5 years after the training data.

**GBRT citation -** It is not clear what software libraries or packages the authors used to train their GBRTs, they should be cited for reproducibility and to give the packaged developers credit for their work.

The library used, scikit-learn, has been now correctly acknowledged and cited in line 135: "We use the GBRT implementation from the scikit-learn library (Pedregosa et al., 2011)".